# Effects of Dietary Alfalfa Meal Supplementation on the Growth Performance, Nutrient Apparent Digestibility, Serum Parameters, and Intestinal Microbiota of Raccoon Dogs (*Nyctereutes procyonoides*)

**DOI:** 10.3390/ani14040623

**Published:** 2024-02-15

**Authors:** Xiaoli Chen, Xiao Li, Danyang Chen, Weigang Zhao, Xiuli Zhang, Weitao Yuan, Huazhe Si, Xuming Deng, Rui Du, Chao Xu

**Affiliations:** 1Institute of Special Animal and Plant Sciences, Chinese Academy of Agricultural Sciences, Changchun 130112, China; chenxiaoli124@163.com (X.C.); lix9805@163.com (X.L.); chendanyang@caas.cn (D.C.); zwg1163@126.com (W.Z.); 15978688053@163.com (W.Y.); 2College of Veterinary Medicine, Jilin University, Changchun 130062, China; xiuli23@jlu.edu.cn (X.Z.); dengxm@jlu.edu.cn (X.D.); 3College of Animal Science and Technology, Jilin Agricultural University, Changchun 130118, China; sihuazhe1989@163.com (H.S.); durui@jlau.edu.cn (R.D.)

**Keywords:** raccoon dog, alfalfa meal, dietary fiber, growth performance, nutrient apparent digestibility, serum parameters, intestinal microbiota

## Abstract

**Simple Summary:**

Raccoon dogs are typically omnivorous and exhibit a diverse dietary range, encompassing small mammals as well as plant matter, which underscores their high adaptability and renders them particularly intriguing subjects for study. Alfalfa meal, renowned for its superior nutritional composition, is employed as a high-quality feed ingredient for ruminants. Diet represents an important determinant influencing the intestinal microbiota. In this study, raccoon dogs were fed diets containing alfalfa meal to scrutinize alterations in their intestinal microbiota. The results showed that supplementation with appropriate alfalfa meal could enhance intestinal health without affecting the growth and overall health of raccoon dogs. This research indicates that raccoon dogs possess a certain tolerance to supplementation with alfalfa meal, thus confirming their ability to tolerate rough feeding.

**Abstract:**

The raccoon dog (*Nyctereutes procyonoides*) is a typical omnivore possessing wide dietary adaptability and tolerance to rough feeding, which may be attributed to its intestinal microbiota. The study aimed to investigate the effect of dietary alfalfa meal levels on the growth performance, nutrient apparent digestibility, serum parameters, and intestinal microbiota of raccoon dogs. Sixty raccoon dogs were randomly divided into four dietary treatments containing 0% (AM0), 5% (AM5), 10% (AM10), and 15% (AM15) alfalfa meal for a 60-day experiment. The results showed that compared to raccoon dogs fed the AM0 diet, those fed the AM5 and AM10 diets had no significant difference in growth performance, while those fed the AM15 diet experienced a significant decrease. Raccoon dogs fed the AM5 diet had no significant effect on the nutrient apparent digestibility. Dietary supplementation with alfalfa meal significantly decreased serum urea levels and increased the antioxidant capacity of raccoon dogs. The intestinal microbiome analysis showed that the richness and diversity of colonic microbiota significantly increased in the AM15 group. With the increase in dietary alfalfa meal levels, the relative abundance of fiber-degrading bacteria in the colon of raccoon dogs, such as *Treponema*, *Phascolarctobacterium*, and *Christensenellaceae R-7 group*, increased. However, the relative abundance of pathogenic bacteria, including *Anaerobiospirillum*, decreased. In conclusion, the inclusion of 5% alfalfa meal in the raccoon dogs’ diet had no effect on growth performance, but it exhibited the potential to improve serum antioxidant capacity and intestinal microbiota. This indicates that raccoon dogs have a certain tolerance to the addition of alfalfa meal in their diet.

## 1. Introduction

The raccoon dog belongs to the order Carnivora and the family Canidae [1]. Its small, plump body and short legs place it at a disadvantage in the realm of carnivores. As a typical omnivore, the characteristics and functions of the raccoon dog’s digestive system are intermediate between those of carnivores and herbivores. The raccoon dog can tolerate roughage, making it suitable for feeding and digesting both animal and plant-based feeds [2]. This distinctive trait is intrinsically tied to their specialized digestive system. The raccoon dog possess a relatively longer intestine compared to other canids, approximately 7.5 times its body length, aiding in the digestion and absorption of plant-derived diets that are more difficult to degrade [3]. The efficient digestion of dietary fiber (DF) is linked to the presence of intestinal microbiota [4,5]. Research has demonstrated that DF intake plays an important role in maintaining intestinal homeostasis and bestows significant beneficial physiological, metabolic, and immunological effects upon the host [6]. Furthermore, different levels of fiber can yield distinct impacts on the intestinal microbiota, leading to the adaptation of varied bacterial communities [7,8]. For instance, in pigs, the consumption of a soybean hull diet led to a reduction in the abundance of *Ruminococcus_1* and *Selenomonas*, while it increased in those fed the sugar beet pulp diet and/or the defatted rice bran diet [9].

DF plays a positive role in promoting the diversity of intestinal microbial communities and maintaining intestinal health. Alfalfa (*Medicago sativa* L.), as a high-quality fiber feed, consists mainly of insoluble dietary fiber (IDF), accounting for more than 90% of the total dietary fiber (TDF) [10,11]. Alfalfa possesses high-fiber properties that stimulate the growth of fiber-degrading bacteria, accelerate intestinal peristalsis, and enhance digestion and absorption in animals [12]. Additionally, alfalfa is rich in proteins, minerals, vitamins, and other essential nutrients, and it contains various bioactive substances [13]. Therefore, there are many studies that have explored the addition of alfalfa to animal diets to regulate intestinal microbiota and improve intestinal health. A previous study found that adding 50% alfalfa meal to Tibetan pigs’ diet increased the abundance of fiber-degrading bacteria (*UCG-005*, *Rikenellaceae_RC9_gut_group*, and *Prevotellaceae_UCG003*) in their hindgut, while reducing the abundance of certain pathogenic bacteria (*Streptococcus*) [14]. Similarly, in a study involving Beijing-you chickens, the addition of 10% alfalfa meal tended to stimulate the proliferation of beneficial bacteria (*Lactobacillus* and *Bacteroides*) and inhibit potential pathogens (*Clostridium*) [15]. These findings underscore the capacity of DF to regulate the intestinal microbiota of organisms.

Until now, numerous studies have examined the impact of alfalfa meal on the growth and intestinal microbiota of livestock and poultry [14,15]. However, no reports have investigated its effects on raccoon dogs. Therefore, this experiment aimed to examine the impact of various levels of alfalfa meal on the growth performance, nutrient apparent digestibility, serum parameters, and intestinal microbiota of raccoon dogs. The primary objectives were to elucidate the role of intestinal microbiota in raccoon dogs’ omnivorous feeding behavior, uncover the capacity of key intestinal microflora in raccoon dogs to digest and absorb DF, and identify the key microbial taxa and functional metabolites that influence raccoon dogs’ digestion of DF.

## 2. Materials and Methods

### 2.1. Animals, Diets and Experimental Design

The feeding experiment was conducted at the fur animal production base of the Institute of Special Animal and Plant Sciences, Chinese Academy of Agricultural Sciences, and the experimental animals were from animals bred in the base that year. Sixty healthy male black raccoon dogs (120 ± 5 days old, initial body weight = 4.57 ± 0.35 kg) were divided into 4 groups, with 15 animals per treatment, in a completely randomized design. Four experimental diets were formulated, consisting of a corn–soybean meal basal diet and three basal diets supplemented with alfalfa meal at the levels of 5%, 10%, and 15% (treatments denoted as AM0, AM5, AM10, and AM15, respectively) (Table 1). None of the experimental diets contained antibiotics or other additives. Before the 60-day experiment began, all animals underwent a 7-day adaptation period to the new diets. Vitamins and minerals were included in all diets to meet or exceed the nutrient requirements of raccoon dogs according to the National Research Council [16]. Throughout the trial period, raccoon dogs were individually caged and provided with access to water ad libitum. They were fed with two equal feedings per day at 07:00 and 14:00, respectively. The animal study was reviewed and approved by the Ethics Committee of the Laboratory Animal Administration of the Institute of Special Animal and Plant Sciences (Approval No. ISAPSAEC-2022-59RD).

### 2.2. Sample Collection and Processing

During the formal experimental period, all raccoon dogs were weighed before the morning feeding on day 0 and 60. Additionally, daily feed supply and residual feed quantity were recorded to calculate the average daily gain (ADG), average daily feed intake (ADFI), and feed-to-gain ratio (F/G). The digestion trail lasted for 3 days, specifically from day 35 to 37 of the trial period. For this trial, seven raccoon dogs with normal food intake and defecation were selected from each group. Using the total collection method, the feces were mixed evenly, sprayed with an appropriate amount of a 10% sulfuric acid solution for nitrogen fixation, dried at 65 °C to a constant weight, powdered through a 0.45 mm (40 mesh) sieve, and preserved at room temperature for chemical analysis. At the end of the feeding experiment, seven raccoon dogs were randomly selected from each group, and 10 mL of blood was collected from the hind limb vein in the morning before feeding using 5 mL vacuum blood collection tubes without any additives. The blood samples were centrifuged at 3500 r/min for 10 min and stored at −80 °C. Afterward, the raccoon dogs were euthanized, and their cecal and colonic contents were collected in sterile frozen tubes and stored in liquid nitrogen immediately. These samples were then transferred to −80 °C for subsequent testing.

### 2.3. Chemical Analysis and Calculation

All chemical compositions of diets and feces were analyzed according to the methods specified by the Association of Analytical Chemists [17]. These analyses included dry matter (DM, method 934.01), crude protein (CP, method 954.01), ether extract (EE, method 920.39), crude fiber (CF, 962.09), neutral detergent fiber (NDF, 2002.04), acid detergent fiber (ADF, method 973.18), and crude ash (method 942.05) in diets, feces, and alfalfa meal. To determine the gross energy (GE) concentration in feed samples, an oxygen bomb calorimeter (IKA-Calorimeter C2000, IKA Company, Staufen, Germany) was used.

### 2.4. Determination of Serum Parameters

The levels of serum biochemical parameters were determined using a Beckman AU480 automatic biochemistry analyzer (Vitalab Selectra E, Spankeren, The Netherlands). The parameters analyzed included serum total protein (TP), albumin (ALB), lactate dehydrogenase (LDH), aspartate aminotransferase (AST), alanine aminotransferase (ALT), alkaline phosphatase (ALP), total cholesterol (T-CHO), triglycerides (TG), high-density lipoprotein cholesterol (HDL-C), low-density lipoprotein cholesterol (LDL-C), urea, and glucose (GLU). For these analyses, commercial colorimetric kits were used, which were purchased from Beijing Zhongsheng Beikong Biochemical Co., LTD. (Beijing, China). Serum TP and ALB concentrations can be used to determine globulin (GLB) concentration (GLB = TP – ALB). The serum total antioxidant capacity (T-AOC), superoxide dismutase (SOD) activity, and malondialdehyde (MDA) content were detected by diagnostic kits (Nanjing Jiancheng Bioengineering Institute, Nanjing, China). Additionally, the concentrations of immunoglobulin A (IgA), immunoglobulin G (IgG), and immunoglobulin M (IgM) in the serum were measured though a double-antibody one-step sandwich enzyme-linked immunosorbent assay (microplate reader, BioTek, Winooski, VT, USA).

### 2.5. Bacterial DNA Extraction, PCR Amplification, and Illumina MiSeq Sequencing

For microbiota analysis, seven cecal and colonic content samples from each group of raccoon dogs were selected, and these samples were sent to Novogene Co., Ltd. (Beijing, China) for sequencing. To extract the total genome DNA from the samples, the CTAB/SDS method was employed. Subsequently, the V3–V4 hypervariable regions of the bacterial 16S rRNA gene were amplified using the polymerase chain reaction (PCR) with primers 341F (5′-CCTAYGGGRBGCASCAG-3′) and 806R (5′-GGACTACNNGGGTATCTAAT-3′). Sequencing libraries were generated with NEBNext^®^ Ultra™ II DNA Library Prep Kit (Cat No. E7645). The quality of the libraries was assessed using the Qubit@ 2.0 Fluorometer (Thermo Scientific, Waltham, MA, USA) and the Agilent Bioanalyzer 2100 system (Santa Clara, CA, USA). Finally, the libraries were subjected to sequencing on an Illumina NovaSeq 6000 platform (San Diego, CA, USA), generating 250 bp paired-end reads for further analysis.

### 2.6. Bio-Informational Analysis

The FLASH software (Version 1.2.11) [18] was utilized to assemble the reads of the samples, generating Raw Tags. Quality control of the Raw Tags was performed using the fastp software (Version 0.20.0). The Tags were compared with the Silva database to identify and remove chimeric constructs, ultimately obtaining the Effective Tags [19]. These Effective Tags were denoised using the DADA2 in the QIIME2 software (Version QIIME2-202006) to obtain initial Amplicon Sequence Variants (ASVs), and ASVs with an abundance less than 5 were eliminated [20]. The QIIME2 software was employed to align the ASVs against a reference database, enabling the assignment of taxonomic information to each ASV. The absolute abundance of ASVs was normalized using a standard of the sequence number corresponding to the sample with the fewest sequences.

The QIIME2 software was used to calculate the Observed_otus, Chao1, Shannon, and Simpson indices. Principal coordinate analysis (PCoA) was conducted based on Bray–Curtis distances. Differences in the bacterial communities among the four groups were analyzed using Adonis functions within the QIIME2 software. The *t*-test was conducted using R software (Version 3.5.3) to assess the significantly different species between each of the two groups at the phylum and genus levels. Furthermore, for identifying the biomarkers between four groups at the genus level, the LEfSe software (Version 1.0) was utilized to perform LEfSe analysis (LDA score threshold was 4).

### 2.7. Determination of Short-Chain Fatty Acids (SCFAs)

The concentration of SCFAs, including acetate, propionate, and butyrate, in the colonic contents was measured by a gas chromatography (GC) system (Agilent Technologies Inc., Santa Clara, CA, USA), as previously described with slight modifications [21]. In brief, approximately 0.2 g of wet digesta was mixed with 6 mL of ultrapure water at 4 °C for 6 h, and then centrifuged at 5000× *g* at 4 °C for 10 min to obtain the supernatant. Metaphosphoric acid (25%, *w*/*v*) was mixed with the extracts at a 1:5 ratio. After centrifugation at 12,000× *g* for 15 min at 4 °C, the supernatant was passed through a 0.22 μm filter membrane and subjected to SCFAs analysis. The samples were analyzed on an Agilent HP5 silica capillary column (30 m × 0.32 mm × 0.32 μm). The temperature program was as follows: the initial temperature was 60 °C, which was then increased at a rate of 10 °C/minute to 170 °C, and further raised by 8 °C/minute to 212 °C. High-purity nitrogen was used as a carrier gas. The injector temperature was 250 °C, and the detector temperature was 270 °C.

### 2.8. Data Statistics and Analysis

The data were analyzed using SPSS 26.0 software, and all values were expressed as mean ± SEM. Prior to data analysis, the Shapiro–Wilk normality test and normal Q-Q plots were used for the normality test. One-way analysis of variance (ANOVA) was conducted to analyze statistical data. To compare the differences among groups, Tukey’s test was utilized. A significance level of *p* < 0.05 was considered statistically significant, while 0.05 < *p* < 0.1 indicated a significant trend. All graphs were generated using GraphPad Prism 9 (GraphPad Software Inc., San Diego, CA, USA). The error bars in all figures represented the standard errors of the mean.

## 3. Results

### 3.1. Effects of Alfalfa Meal Supplementation on Growth Performance of Raccoon Dogs

The effect of diets containing different levels of alfalfa meal on the growth performance of raccoon dogs is presented in Table 2. Upon the inclusion of alfalfa meal in the diet, factors such as FBW, ADG, and ADFI exhibited a decrease, while F/G increased. When compared with the AM0 group, there were no significant differences in FBW, ADG, and F/G in the AM5 and AM10 groups (*p* > 0.05), while the AM15 group witnessed a significant decline in FBW, ADG, and ADFI, alongside a notable increase in F/G (*p* < 0.05). When comparing the AM10 group to the AM5 group, there were no significant differences in FBW, ADG, and F/G (*p* > 0.05), while the AM15 group recorded a significant decrease in FBW, ADG, and ADFI, coupled with a significant increase in F/G (*p* < 0.05). Furthermore, there was no significant difference in FBW between the AM10 group and the AM15 group (*p* > 0.05). In conclusion, the results indicate that supplementing raccoon dogs’ diets with 5% and 10% of alfalfa meal did not significantly affect their growth performance. However, when raccoon dogs were supplemented with 15% alfalfa meal, there was a notable decrease in their growth performance.

### 3.2. Effects of Alfalfa Meal Supplementation on Nutrient Apparent Digestibility of Raccoon Dogs

The nutrient apparent digestibility of raccoon dogs fed with different levels of alfalfa meal was evaluated (Table 3). As the dietary level of alfalfa meal increased, the digestibility of DM and ADF decreased. However, there were no significant differences in CP, EE, and NDF digestibility among all the groups (*p* > 0.05). Specifically, the DM intake in the AM15 group was significantly lower than that in the other three groups, the DM output in the AM5 group was significantly lower than that in the AM10 and AM15 groups (*p* < 0.05). In comparison to the AM0 group, the digestibility of DM and ADF in the AM5 group showed no significant differences (*p* > 0.05), while the digestibility of DM and ADF in the AM10 group significantly decreased, and the digestibility of DM in the AM15 group also significantly decreased (*p* < 0.05). When compared to the AM5 group, DM digestibility in the AM10 and AM15 groups significantly decreased (*p* < 0.05). Furthermore, the digestibility of DM, CP, NDF, and ADF in raccoon dogs from the AM15 group was higher than that in the AM10 group (*p* > 0.05). In summary, the results indicate that a dietary inclusion of 5% alfalfa meal had no significant effect on the nutrient apparent digestibility of raccoon dogs. However, a 10% inclusion of alfalfa meal significantly decreased the digestibility of DM and ADF, and a 15% inclusion of alfalfa meal significantly decreased the digestibility of DM in raccoon dogs.

### 3.3. Effects of Alfalfa Meal Supplementation on Serum Parameters of Raccoon Dogs

Serum parameters are closely linked to nutrient metabolism and the occurrence of diseases in animals. It is important to note that any changes in diet composition can result in alterations to serum parameters. The effects of different levels of dietary alfalfa meal on the serum biochemical parameters of raccoon dogs are presented in Table 4. The inclusion of different proportions of alfalfa meal did not significantly affect the serum TP, ALB, and GLB levels in raccoon dogs (*p* > 0.05). Similarly, there were no significant differences in LDH and ALP activities among the four groups (*p* > 0.05). The inclusion of different proportions of alfalfa meal did result in reduced serum AST and ALT activities (*p* > 0.05). As the dietary alfalfa meal level increased, the serum urea level in raccoon dogs decreased. Among them, the serum urea levels in the AM10 and AM15 groups were significantly lower than those in the AM0 group (*p* < 0.05). In contrast, changes in the dietary alfalfa meal level had no significant effects on the serum lipid metabolism indices (T-CHO, TG, HDL-C, and LDL-C) or GLU content in raccoon dogs (*p* > 0.05).

The serum antioxidant indices of raccoon dogs were assessed following the consumption of diets with varying levels of alfalfa meal (Table 5). No significant difference in T-AOC was observed among the groups. With an increase in dietary alfalfa meal levels, the serum SOD activity in raccoon dogs showed a significant increase, while the MDA content significantly decreased (*p* < 0.05). As shown in Appendix A, it is evident that incorporating alfalfa meal into the diet had no significant impact on the serum immune indices (IgA, IgG, and IgM) of raccoon dogs (*p* > 0.05).

### 3.4. Effects of Alfalfa Meal Supplementation on Microbial Diversity of Raccoon Dogs

The utilization of fiber in the hindgut of raccoon dogs primarily depends on microbial fermentation. Consequently, we examined the diversity and microbial community composition in the cecum and colon of raccoon dogs. The coverage index for all samples exceeded 0.995, indicating that the sequencing depth was adequate (Appendix A). To evaluate the alpha diversity of intestinal microbiota, we utilized richness indices (Observed_otus and Chao1) and diversity indices (Shannon and Simpson). As shown in Figure 1, dietary levels of alfalfa meal had no significant effect on the cecal microbiota richness and diversity in raccoon dogs (*p* > 0.05). However, in the colon of raccoon dogs, the AM15 group displayed a significantly higher number of OTUs, Chao1 index, and Shannon index when compared to the AM0 and AM10 groups (*p* < 0.05, Figure 1A–C). There were no significant differences in the Simpson index of the colonic microbiota among the diets with different levels of alfalfa meal (*p* > 0.05, Figure 1D).

Subsequently, we explored the influence of various dietary levels of alfalfa meal on the intestinal microbiota community structure. The PCoA plots, based on Bray–Curtis distance, portrayed the microbial community composition of raccoon dogs fed with different diets (Figure 2, Table 6). The PCoA results demonstrated that with regard to both species presence and species abundance, the intestinal microbiota composition underwent alterations with the increase in dietary alfalfa meal levels. In the cecum of raccoon dogs, the intestinal microbiota composition in the AM15 group significantly differed from that in the AM0 (Adonis: *R*^2^ = 0.1796, *p* = 0.020), AM5 (Adonis: *R*^2^ = 0.1571, *p* = 0.020), and AM10 (Adonis: *R*^2^ = 0.1560, *p* = 0.028) groups, respectively (Figure 2A). Similarly, in the colon, significant differences were observed in the composition of gut microbiota among the four dietary groups (Figure 2B). The findings underscore that dietary alfalfa meal can impact the intestinal microbial structure of raccoon dogs. Notably, when the level of alfalfa meal supplementation reached 15%, the richness and diversity of the colonic microbiota in raccoon dogs were significantly increased.

### 3.5. Composition and Differences of Intestinal Microbiota in Raccoon Dogs

At the phylum level, Bacteroidota and Firmicutes were the dominant phyla (Figure 3A). Bacteroidota were the first dominant phylum in every group, accounting for 56.26% to 79.37%. Firmicutes accounted for 7.97% to 26.63% in each group, making it the second dominant phylum, followed by Fusobacteriota, Proteobacteria, Spirochaetota, and Actinobacteriota (Appendix A). The relative abundance of Bacteroidota, Firmicutes, and Spirochaetota was significantly influenced by the level of dietary alfalfa meal (*p* < 0.05) (Appendix A). In the cecum of raccoon dogs, compared to the AM0 group, the relative abundance of Bacteroidota significantly decreased in the AM5 group, while the relative abundance of Spirochaetota and Campilobacterota significantly increased in the AM15 group (*p* < 0.05). In comparison to the AM5 group, the relative abundance of Bacteroidota in the AM10 group and Spirochaetota and Campilobacterota in the AM15 group significantly increased (*p* < 0.05). Furthermore, compared to the AM10 group, the relative abundance of Firmicutes in the AM15 group significantly decreased, while the relative abundance of Spirochaetota and Campilobacterota significantly increased (*p* < 0.05). In the colon, compared to the AM0 group, the relative abundance of Bacteroidota significantly decreased in the AM5 group, while the relative abundance of Fusobacteriota and Proteobacteria significantly increased (*p* < 0.05). In the AM10 and AM15 groups, the relative abundance of Bacteroidota significantly decreased, while the relative abundance of Firmicutes and Spirochaetota significantly increased (*p* < 0.05). Moreover, when compared to the AM5 group, the relative abundance of Bacteroidota and Spirochaetota in the AM10 group significantly increased, while the relative abundance of Firmicutes and Spirochaetota in the AM15 group significantly increased, and the relative abundance of Proteobacteria significantly decreased (*p* < 0.05). When compared to the AM10 group, the relative abundance of Bacteroidota in the AM15 group significantly decreased, while the relative abundance of Firmicutes significantly increased (*p* < 0.05).

Furthermore, we conducted an analysis of the intestinal microbiota at the genus level in raccoon dogs. According to the definition of dominant species, with a relative abundance of more than 10%, *Prevotella* emerged as the first dominant genus in all groups, accounting for 34.43% to 56.62%, followed by *Alloprevotella*, *Fusobacterium*, *Bacteroides*, *Treponema*, and others (Figure 3B, Appendix A). Additionally, the relative abundance of these bacterial genera was similarly significantly influenced by the level of dietary alfalfa meal (Appendix A). In the cecum of raccoon dogs, when compared to the AM0 group, the relative abundance of *Prevotella* and *Alloprevotella* significantly decreased in the AM5 group, while the relative abundance of *Alloprevotella* and *Anaerobiospirillum* in the AM10 group significantly decreased (*p* < 0.05). The relative abundance of *Alloprevotella* and *Anaerobiospirillum* in the AM15 group significantly decreased, whereas that of *Treponema* significantly increased (*p* < 0.05). When compared to the AM5 group, the relative abundance of *Prevotella* in the AM10 group significantly increased, while the relative abundance of *Treponema* in the AM15 group significantly increased. Additionally, the relative abundance of *Anaerobiospirillum* significantly decreased (*p* < 0.05). In comparison to the AM10 group, the relative abundance of *Treponema* in the AM15 group significantly increased, while the relative abundance of *Streptococcus* significantly decreased (*p* < 0.05). In the colon, compared to the AM0 group, the relative abundance of *Prevotella* and *Alloprevotella* in the AM5 group significantly decreased, while the relative abundance of *Fusobacterium* significantly increased (*p* < 0.05). In the AM10 group, the relative abundance of *Alloprevotella* significantly decreased, while the relative abundance of *Treponema* significantly increased (*p* < 0.05). In the AM15 group, the relative abundance of *Prevotella*, *Alloprevotella*, and *Anaerobiospirillum* significantly decreased, while the relative abundance of *Treponema* significantly increased (*p* < 0.05). When compared to the AM5 group, the relative abundance of *Prevotella* and *Treponema* in the AM10 group significantly increased (*p* < 0.05). The relative abundance of *Treponema*, *Clostridia_UCG-014*, *Prevotellaceae_UCG-003*, *Rikenellaceae_RC9_gut_group*, *UCG-005*, and *Ruminococcus* in the AM15 group significantly increased (*p* < 0.05). The relative abundance of *Prevotella* was significantly decreased in the AM15 group compared to the AM10 group (*p* < 0.05).

To more accurately identify biomarkers among the four groups, we employed LEfSe multilevel species difference discriminant analysis, which helped to determine the key players with significant effects between the groups, using an LDA score threshold of 4.0 (Figure 4). In the cecum of raccoon dogs, LEfSe analysis revealed 15 biomarkers. Among these, the AM0 group had two taxa, the AM5 group had three taxa, the AM10 group had four taxa, and the AM15 group had six taxa. *Alloprevotella* (LDA = 4.336, *p* = 0.044) was found to be predominant in the AM0 group, while *Bacteroides* (LDA = 4.232, *p* = 0.018) was predominant in the AM5 group. The AM10 group showed a predominance of *Prevotella* (LDA = 4.773, *p* = 0.029) and *Streptococcus* (LDA = 4.113, *p* = 0.007). Moreover, *Treponema* (LDA = 4.428, *p* = 0.021) was identified as the biomarker of the AM15 group. Moving on to the colon, LEfSe analysis identified 31 biomarkers. The AM0 group comprised seven taxa, the AM5 group had ten taxa, the AM10 group had one taxon, and the AM15 group had thirteen taxa. *Prevotella* (LDA = 4.981, *p* = 0.005) and *Alloprevotella* (LDA = 4.483, *p* = 0.006) were found to be predominant in the AM0 group, while *Fusobacterium* (LDA = 4.715, *p* = 0.019) was predominant in the AM5 group. *Anaerobiospirillum* (LDA = 4.066, *p* = 0.004) and *Treponema* (LDA = 4.342, *p* = 0.004) were identified as the biomarkers for the AM10 and AM15 groups, respectively.

### 3.6. Correlation Analysis between Significantly Different Colon Microbiota and Growth Performance, Nutrient Apparent Digestibility, and Serum Parameters

As shown in Figure 5, further correlation analysis of colonic microbiota and growth performance, nutrient apparent digestibility, and serum parameters showed that *Cetobacterium* was significantly positively correlated with ADG, while *Prevotellaceae_UCG-003*, *Phascolarctobacterium*, *UCG-005*, *Ruminococcus*, *Lactobacillus*, *Anaerovibrio*, and *Christensenellaceae R-7 group* were significantly negatively correlated with it. *Alloprevotella* and *Cetobacterium* were significantly positively correlated with DM digestibility, while *Prevotellaceae_UCG-003*, *UCG-005*, *Ruminococcus*, *Anaerovibrio*, *Oribacterium*, *Eubacterium_siraeum_group*, and *Oscillospira* were significantly negatively correlated with it. *Phascolarctobacterium*, *Eubacterium_coprostanoligenes_group*, and *Desulfovibrio* were significantly positively correlated with SOD, while *Coprococcus* was significantly negatively correlated with it. The results showed that there was a certain correlation between colonic microbiota and growth performance, nutrient apparent digestibility, and serum parameters.

### 3.7. Concentrations of SCFA in Colonic Contents

The concentration of SCFA in the colonic contents of raccoon dogs fed diets with different alfalfa meal levels were further studied (Figure 6). It was found that dietary alfalfa meal had no significant effects on the concentrations of acetate, propionate, butyrate, and total SCFA (*p* > 0.05).

## 4. Discussion

Alfalfa and other forages were primarily used to feed ruminants. However, only a few studies have investigated the inclusion of alfalfa meal in the diets of monogastric livestock due to the digestive physiological structure of monogastric animals, which limits their efficient utilization of these plant-based raw materials [22]. Raccoon dogs are omnivorous canids, suggesting that they might possess the ability to digest and utilize such materials to some extent [23]. Consequently, we conducted an assessment of the effects of dietary alfalfa meal on the growth performance, nutrient utilization, serum parameters, and intestinal microbiota of raccoon dogs.

Our findings revealed that an elevated dietary alfalfa meal level led to a decrease in the FBW, ADG, and ADFI of raccoon dogs, while leading to an increase in the F/G. This implies that high levels of alfalfa meal hindered the growth of raccoon dogs. This phenomenon could be attributed to the increase in dietary crude fiber content, which accelerated the passage of digestate through the digestive tract, consequently reducing gastrointestinal transit time and ultimately leading to a decline in growth performance [24]. In the present study, the optimal dietary alfalfa meal level for raccoon dogs was determined to be 5%, because the growth performance of the AM5 group was similar to that of the AM0 group. This may be attributed to the similar levels of CF, NDF, and ADF in the two diets, resulting in little effects on nutrient digestibility [25]. DF can influence dietary palatability and animal feed intake [26]. Notably, the increase in dietary alfalfa meal was found to significantly reduce ADFI, indicating that higher DF levels resulting from increased alfalfa meal content decreased dietary palatability, consequently reducing the dietary intake by raccoon dogs.

Nutrient apparent digestibility is closely related to the growth of animals. Alfalfa meal is abundant in DF, of which IDF accounts for more than 90% of the TDF. DF serves as an anti-nutrient factor, which can decrease dietary energy levels and hinder nutrient digestion, potentially leading to decreased growth performance in some instances [27]. Actually, the digestibility of nutrients is intricately linked with the time digesta remains within the digestive system [28]. The present study found that as the levels of alfalfa meal (5%, 10%, and 15%) increased, there was a reduction in the apparent digestibility of DM and ADF in raccoon dogs. This may be because the increased IDF level in the diet reduces the viscosity of the digesta and accelerates the fermentation of the digesta in the posterior intestine. Interestingly, the nutrient apparent digestibility in the AM15 group was higher than that of the AM10 group, while it was similar with the AM5 group. This may be attributed to the similar amino acid composition of diets in the AM5 and AM15 groups, or that the balance of amino acids in the AM15 group’s diet was achieved [29]. However, the specific reason needs further research.

Serum parameters can reflect the metabolic and health status of organisms. Urea is the main nitrogenous end product of protein metabolism and decomposition in mammals, and its level in serum correlates with the quantity and quality of protein and amino acids in the diet. Lower serum urea levels indicate higher dietary protein quality [30]. In this study, the decreased serum urea level was observed after supplementing with alfalfa meal, implying that the addition of alfalfa meal improved the protein utilization rate of raccoon dogs and increased nitrogen deposition. T-AOC can directly reflect the activity of antioxidant enzymes and indirectly reflect the degree of lipid peroxidation damage [31]. SOD plays an important role in the removal of reactive oxygen species [32]. MDA is an end product of lipid peroxidation in biofilm, and its level can reflect the degree of lipid peroxidation [33]. Previous study has shown that adding alfalfa to the diet enhances the body’s antioxidant capacity [34]. We also obtained similar results since our study found that the increase in the alfalfa meal level led to an increased trend in T-AOC activity, increased SOD activity, and decreased MDA content in the serum. This may be related to the various bioactive components contained in alfalfa meal, such as alfalfa saponins, flavonoids, polysaccharides, alkaloids, and DF [10,11]. Flavonoids isolated from alfalfa have been shown to enhance DPPH free-radical-scavenging activity and antioxidant activity [35]. Alfalfa polysaccharides can improve the body’s antioxidant properties by scavenging hydroxyl groups in the Fe^2+^/H_2_O_2_ system. Furthermore, DF in alfalfa can disrupt the absorption and synthesis of bile acids, impacting bile acid metabolism. This can increase the excretion of fat in the feces, causing a decrease in serum lipid levels and tissue lipid storage, thereby mitigating lipid peroxidation in the body [36].

The intestinal tract contains a large number and a wide variety of microorganisms. These intestinal microorganisms play an important role in regulating host nutrient metabolism and immune function [37]. In order to ascertain the potential correlation between the aforementioned alterations and intestinal microbiota, an analysis was conducted on the alpha and beta diversity of the cecum and colon microbiota in raccoon dogs. We found that the inclusion of alfalfa meal in the diet increased the richness and diversity of colonic microbiota. Additionally, there were also differences in the response of the colonic microbiota to different alfalfa meal levels.

It was previously found that the Bacteroidota and Firmicutes are predominant intestinal microbiota in the raccoon dogs’ intestine [38], and similar results were obtained in our study. Our results showed that dietary supplementation of alfalfa meal mainly affected the colonic microbial diversity of raccoon dogs. Therefore, this work mainly focused on the differential flora present in the colon. Our results revealed that the relative abundance of *Prevotella* and *Alloprevotella*, which belong to the Bacteroidota phylum, was higher in the control group than in the group supplement with alfalfa meal, suggesting that alfalfa meal may have an adverse impact on Bacteroidota growth. This probably explains the higher intestinal alpha diversity but lower growth performance in the AM15 group compared to the AM0 group. Spearman correlation analysis revealed that the digestibility of DM has a significantly positive correlation with the relative abundance of *Alloprevotella*. Firmicutes contains a significant number of cellulose-degrading bacteria and, as a result, has a higher fiber-degrading capacity [39]. Research has found that Firmicutes significantly increased with fiber inclusion in canine diets [40]. Similar results were obtained in our study, where the addition of alfalfa meal to the raccoon dogs’ diet significantly increased the relative abundance of Firmicutes. The bacteria *Phascolarctobacterium*, *UCG-005*, and *Christensenellaceae R-7 group* in the Firmicutes actively contribute to the decomposition of fibrous substances. *Phascolarctobacterium* is associated with SCFA production [41]. *UCG-005* significantly correlates with acetate and total SCFA concentrations [42]. *Christensenellaceae group R-7*, which belongs to the Christensenellaceae family, diversely ferments sugars, yielding acetate and butyrate [43]. Our results found that as the dietary intake of alfalfa meal increased, the relative abundance of these fiber-decomposing bacteria increased in the colon of raccoon dogs, aligning with previous findings [44,45]. It has been reported that an increased DF content could increase Firmicutes and reduce Bacteroidota abundance, resulting in an increased Firmicutes/Bacteroidota (F/B) ratio [40]. Additionally, research conducted on mice has also demonstrated that those fed with a high-fiber diet display a higher F/B ratio compared to those on a low-fiber diet [46]. Therefore, we speculated that the decline in Bacteroidota might arise from competition for nutrients or energy with the faster-growing fiber-degrading bacteria in Firmicutes [47]. *Clostridia_UCG-014* negatively correlates with liver function indicators, particularly ALT levels [48]. We found that the serum ALT levels in raccoon dogs gradually decreased with the increase in alfalfa meal intake. The AM15 group showed the lowest ALT levels, correlating with the highest *Clostridia_UCG-014* abundance, suggesting its positive impact on the liver health of raccoon dogs. Research has indicated that supplementation with non-digestible fiber in the diet, such as fructooligosaccharides, can elevate *Lactobacillus* abundance and regulate the flora [49]. Our results found that raccoon dogs in the AM15 group exhibited higher *Lactobacillus* in the colon compared to the AM0 group, illustrating that DF supplementation can enhance the relative abundance of *Lactobacillus*.

Fusobacteriota, a cluster of obligate anaerobic Gram-negative bacilli, is known to be a normal component present in the intestinal microbiota [44]. A recent study found that an increase in Fusobacteriota was associated with diarrhea in newborn piglets [50]. However, a reduction in the abundance of Fusobacteriota was observed in dogs with gastrointestinal ailments [51]. This may indicate that the function of Fusobacteriota is species specific. Our results found a significant increase in the relative abundance of Fusobacteriota in the colon following the supplementation of 5% alfalfa meal in the raccoon dogs’ diet, so we speculate that this may have a beneficial effect on its intestinal health. On the other hand, a lower diet quality has been linked to increased Fusobacteriota and *Fusobacterium* abundance [52]. In the present study, the supplementation of alfalfa meal led to reduced dietary palatability, potentially contributing to the higher relative abundance of Fusobacteriota and *Fusobacterium*. Spearman correlation analysis indicated a significant negative correlation between the abundance of *Fusobacterium* and serum MDA content. This suggests that as the dietary levels of alfalfa meal increased, the relative abundance of *Fusobacterium* increased, leading to an enhancement in the antioxidant capacity of raccoon dogs. *Anaerobiospirillum* is one of the most abundant genera of Proteobacteria in canines. It can serve as an infectious pathogen causing intestinal infections [53]. The relative abundance of *Anaerobiospirillum* in the AM15 group was significantly lower than in the AM0 group, indicating that the alfalfa meal supplementation could reduce the population of pathogenic gut bacteria, thus favoring the animals’ development and overall health status. Spirochaetota plays an important role in composite fiber degradation [54]. *Treponema*, as a member of Spirochaetota, demonstrates an affinity for a high-fiber diet [55]. Although *Treponema* may not directly utilize fiber, it assists other cellulolytic bacteria in cellulose fermentation [56]. The result of this experiment was consistent with the prior research, indicating that alfalfa supplementation resulted in increased *Treponema* abundance [57].

SCFAs are important metabolites of hindgut microbiota fermentation and the final products of DF fermentation [58]. The three primary components, acetate, propionate, and butyrate, constitute over 95% of the total SCFA concentrations [59]. In the present study, the dietary IDF content increased with the increase in alfalfa meal levels. The concentrations of acetate, propionate, butyrate, and total SCFA in the alfalfa meal supplemented group increased in comparison to the control group, although the differences were not significant. This finding was inconsistent with other studies [14,60] on the effects of alfalfa meal on SCFA concentrations in the colonic digesta of animals. Actually, the majority of DF in alfalfa meal is composed of IDF, which may hinder hindgut fermentation, resulting in lower SCFA [61]. In addition, the intestinal fermentation capacity for DF varies among different animals, which may lead to a discrepancy in SCFA concentrations.

## 5. Conclusions

In summary, the present study demonstrated that dietary supplementation of alfalfa meal may influence the growth performance and other parameters of raccoon dogs by modulating the intestinal microbiota. The results revealed that the inclusion of alfalfa meal improved the richness and diversity of the colonic microbiota in raccoon dogs. Moreover, it increased the relative abundance of fiber-degrading bacteria and led to a reduction in pathogenic bacteria, promoting the fermentation of IDF present in alfalfa meal and contributing to the maintenance of the intestinal microecological balance. Based on the present results, a diet containing 5% alfalfa meal had no impact on the growth performance of raccoon dogs. However, it improved serum antioxidant capacity and promoted improvements in the intestinal microbiota. These findings suggest that raccoon dogs possess a certain level of tolerance towards alfalfa meal.

## Figures and Tables

**Figure 1 animals-14-00623-f001:**
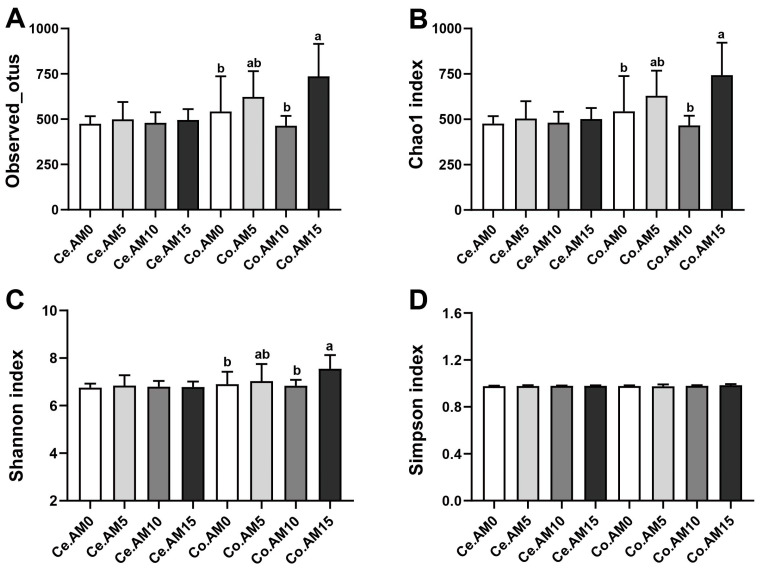
Differences in alpha diversity of cecal and colonic microbiota among the four groups (*n* = 7). Observed_otus (**A**) and Chao1 (**B**) are community richness indices. Shannon (**C**) and Simpson (**D**) are community diversity indices. Ce = cecum, Co = colon, AM = alfalfa meal. ^a,b^ values with different superscripts are significantly different (*p* < 0.05).

**Figure 2 animals-14-00623-f002:**
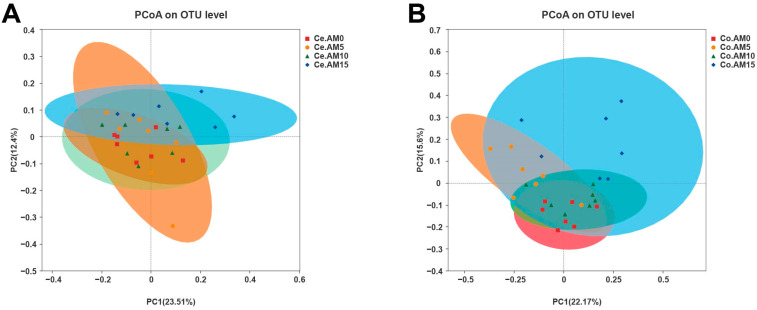
Principal coordinate analysis (PCoA) based on Bray-Curtis distance with Adonis test of the intestinal microbiota community in cecum (**A**) and colon (**B**) (*n* = 7). Ce = cecum, Co = colon, AM = alfalfa meal.

**Figure 3 animals-14-00623-f003:**
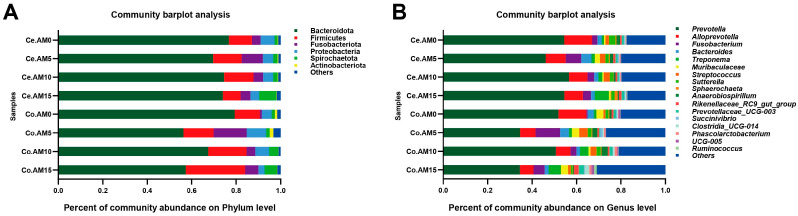
The composition of intestinal microbiota communities at the phylum (**A**) and genus (**B**) level of raccoon dogs (*n* = 7). Ce = cecum, Co = colon, AM = alfalfa meal.

**Figure 4 animals-14-00623-f004:**
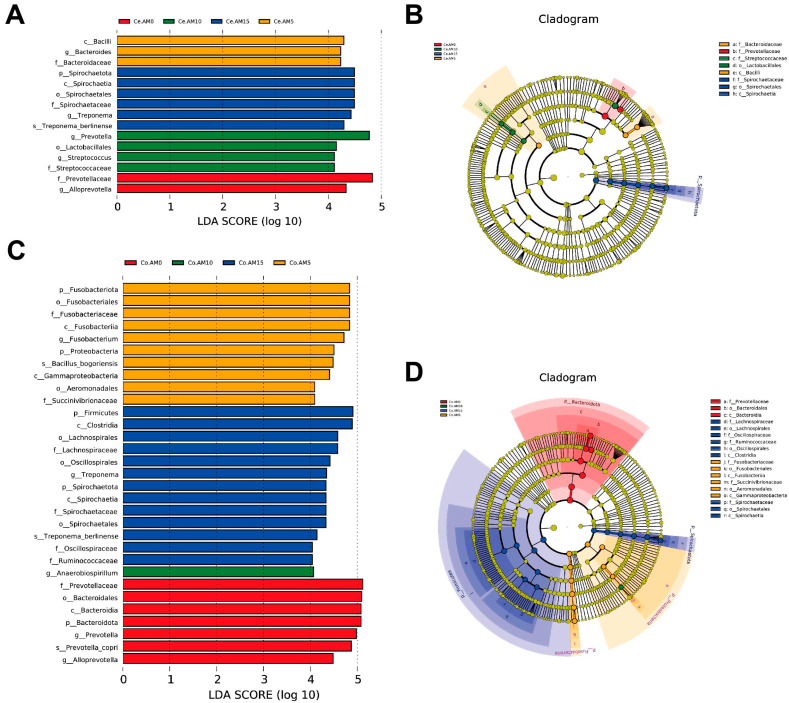
LEfSe map of species differences among the four groups at the genus level (*n* = 7). Highly significant bacteria with LDA score > 4 (**A**,**C**), LEfSe cladogram (**B**,**D**). Ce = cecum, Co = colon, AM = alfalfa meal.

**Figure 5 animals-14-00623-f005:**
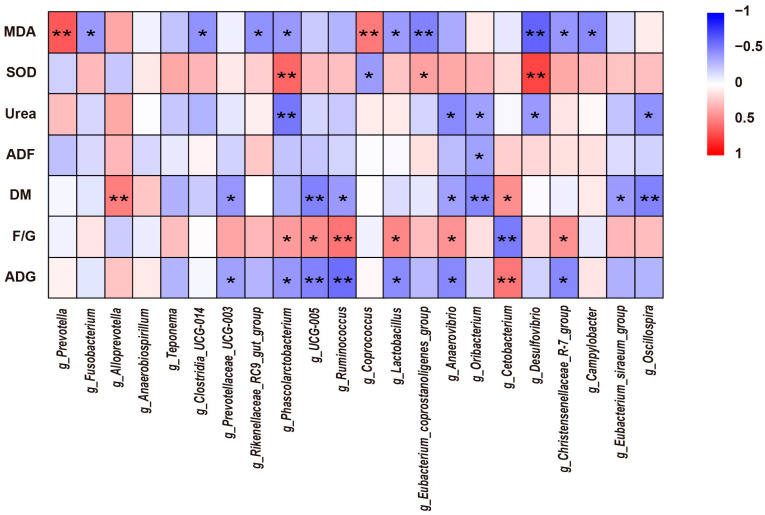
Spearman’s correlation heat map between significantly different colonic microbiota and significantly different growth performance, nutrient apparent digestibility, and serum parameters for different levels of alfalfa meal treatment. ADG = average daily gain; F/G = feed–gain weight ratio; DM = dry matter; ADF = acid detergent fiber; SOD = superoxide dismutase; MDA = malondialdehyde. Significance and correlation coefficient was analyzed by Spearman’s correlation analysis. * *p* < 0.05; ** *p* < 0.01.

**Figure 6 animals-14-00623-f006:**
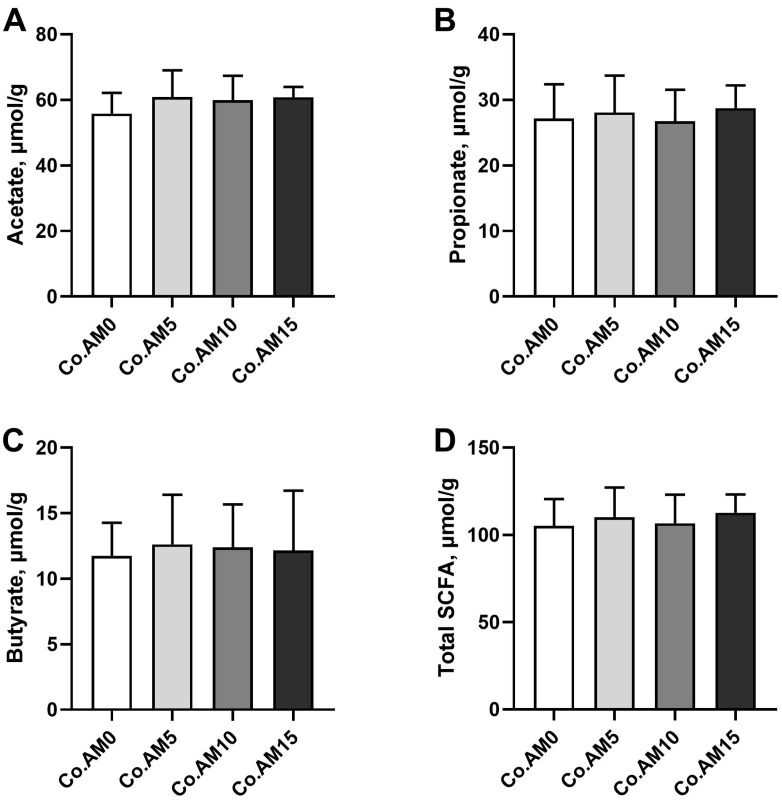
SCFA concentrations (μmol/g wet digesta) measured in the colon of raccoon dogs fed diets with different levels of alfalfa meal (*n* = 7). Concentrations of acetate (**A**), propionate (**B**), butyrate (**C**), and total SCFA (**D**). Co = colon, AM = alfalfa meal.

**Table 1 animals-14-00623-t001:** Ingredients and nutrient composition of the experimental diets (as-fed basis, %).

Item	AM0	AM5	AM10	AM15
Ingredients				
Extruded corn	46.50	46.50	45.50	43.50
Soybean meal	15.00	14.00	14.00	13.00
Alfalfa meal ^1^	0.00	5.00	10.00	15.00
Wheat bran	4.00	2.00	0.00	0.00
Corn germ meal	6.00	6.00	6.00	6.00
Corn skin	6.00	4.00	2.00	0.00
Meat and bone meal	5.00	5.00	5.00	5.00
Fish meal	2.50	2.50	2.50	2.50
Chicken meal	2.00	2.00	2.00	2.00
Plasma protein powder	1.00	1.00	1.00	1.00
Chicken oil	4.00	4.00	4.00	4.00
Glucose	1.50	1.50	1.50	1.50
Milk powder	2.50	2.50	2.50	2.50
Limestone	0.30	0.30	0.30	0.30
Dicalcium phosphate	0.60	0.60	0.60	0.60
L-Lysine	0.40	0.40	0.40	0.40
DL-Methionine	0.30	0.30	0.30	0.30
Sodium chloride	0.40	0.40	0.40	0.40
Vitamin–mineral premix ^2^	2.00	2.00	2.00	2.00
Total	100.00	100.00	100.00	100.00
Nutrient and energy composition ^3^				
Dry matter	92.88	92.96	93.19	92.61
Crude protein	19.90	20.22	20.07	20.68
Ether extract	7.62	7.31	7.52	7.36
Crude fiber	7.76	9.21	11.04	12.21
Neutral detergent fiber	14.84	15.72	19.05	23.17
Acid detergent fiber	5.28	7.25	9.01	10.65
Crude ash	6.17	6.60	6.80	7.20
Calcium	1.13	1.24	1.28	1.38
Total phosphorus	1.38	1.37	1.31	1.24
Gross energy, MJ/kg	18.99	18.90	18.95	18.44

^1^ Alfalfa meal: 90.52% dry matter, 15.53% crude protein, 1.19% ether extract, 35.98% crude fiber, 56.69% neutral detergent fiber, 44.50% acid detergent fiber, 10.29% crude ash. ^2^ This vitamin–mineral premix supplied per kilogram diet was as follows: vitamin A, 6250 IU; vitamin D_3_, 1000 IU; vitamin E, 60 IU; vitamin K_3_ 2 mg; thiamine 12.5 mg; riboflavin, 9.0 mg; pyridoxine, 7.5 mg; vitamin B_12_, 0.0225 mg; D-pantothenic acid, 17.5 mg; niacin, 25 mg; folic acid, 0.5 mg; biotin, 0.1 mg; vitamin C, 62.5 IU; Cu (as copper sulfate), 18 mg; Fe (as iron sulfate), 96 mg; I (as potassium iodate), 1.44 mg; Mn (as manganese sulfate), 48 mg; Se (as sodium selenite), 0.3 mg; Zn (as zinc oxide), 78 mg; Co (as cobalt sulfate), 0.24 mg; choline chloride, 900 mg. ^3^ Nutrient and energy composition were measured values.

**Table 2 animals-14-00623-t002:** Effects of dietary alfalfa meal levels on growth performance of raccoon dogs (*n* = 15).

Item	AM0	AM5	AM10	AM15	SEM	*p*-Value
IBW, kg	4.60	4.60	4.51	4.57	0.05	0.879
FBW, kg	7.49 ^a^	7.44 ^a^	7.20 ^ab^	6.96 ^b^	0.06	0.004
ADG, g/d	48.11 ^a^	47.33 ^a^	44.83 ^a^	39.83 ^b^	0.71	<0.001
ADFI, g/d	308.01 ^a^	307.08 ^b^	306.10 ^c^	305.49 ^d^	0.13	<0.001
F/G	6.47 ^b^	6.53 ^b^	6.87 ^b^	7.80 ^a^	0.11	<0.001

IBW = initial body weight; FBW = final body weight; ADG = average daily gain; ADFI = average daily feed intake; F/G = feed-to-gain ratio. ^a,b,c,d^ with a row, values with different superscripts are significantly different (*p* < 0.05).

**Table 3 animals-14-00623-t003:** Effects of dietary alfalfa meal levels on nutrient apparent digestibility of raccoon dogs (*n* = 7).

Item	AM0	AM5	AM10	AM15	SEM	*p*-Value
DM intake, g	882.75 ^a^	791.83 ^b^	792.96 ^b^	788.74 ^c^	7.65	<0.001
DM output, g	249.34 ^ab^	230.05 ^b^	259.01 ^a^	255.20 ^a^	4.23	0.064
DM digestibility, %	71.76 ^a^	70.95 ^a^	67.34 ^b^	67.64 ^b^	0.58	0.004
CP digestibility, %	68.93	69.41	67.04	68.85	0.51	0.391
EE digestibility, %	87.73	86.50	83.06	82.80	0.83	0.078
NDF digestibility, %	64.44	62.23	62.78	66.00	0.70	0.219
ADF digestibility, %	42.05 ^a^	40.32 ^ab^	31.41 ^b^	38.35 ^ab^	1.47	0.045

DM = dry matter; CP = crude protein; EE = ether extract; NDF = neutral detergent fiber; ADF = acid detergent fiber. ^a,b,c^ with a row, values with different superscripts are significantly different (*p* < 0.05).

**Table 4 animals-14-00623-t004:** Effects of dietary alfalfa meal levels on serum biochemical parameters of raccoon dogs (*n* = 7).

Item	AM0	AM5	AM10	AM15	SEM	*p*-Value
TP, g/L	58.53	57.39	56.78	57.08	0.90	0.920
ALB, g/L	29.31	27.99	28.06	27.59	0.56	0.742
GLB g/L	29.22	29.39	28.72	29.50	1.13	0.996
LDH, U/L	295.25	302.83	293.58	293.48	5.10	0.914
AST, U/L	56.58	55.28	51.30	51.48	1.20	0.303
ALT, U/L	18.82	17.01	16.91	16.26	0.74	0.669
ALP, U/L	23.44	22.66	21.23	21.47	0.63	0.591
Urea, mmol/L	9.99 ^a^	8.26 ^ab^	6.70 ^b^	7.65 ^b^	0.31	<0.001
T-CHO, mmol/L	2.92	2.54	2.68	2.57	0.10	0.526
TG, mmol/L	0.38	0.36	0.36	0.35	0.01	0.869
HDL-C, mmol/L	2.20	1.94	1.90	1.98	0.07	0.535
LDL-C, mmol/L	0.10	0.09	0.10	0.09	0.01	0.745
GLU, mmol/L	2.68	2.48	2.67	2.65	0.07	0.763

TP = total protein; ALB = albumin; GLB = globulin; LDH = lactate dehydrogenase; AST = aspartate aminotransferase; ALT = alanine aminotransferase; ALP = alkaline phosphatase; T-CHO = total cholesterol; TG = triglycerides; HDL-C = high-density lipoprotein cholesterol; LDL-C = low-density lipoprotein cholesterol; GLU = glucose. ^a,b^ with a row, values with different superscripts are significantly different (*p* < 0.05).

**Table 5 animals-14-00623-t005:** Effects of dietary alfalfa meal levels on serum antioxidant indices of raccoon dogs (*n* = 7).

Item	AM0	AM5	AM10	AM15	SEM	*p*-Value
T-AOC, mM	0.34	0.36	0.37	0.37	0.01	0.543
SOD activity, U/mL	10.17 ^b^	12.32 ^a^	12.31 ^a^	12.71 ^a^	0.32	0.012
MDA content, nmol/mL	5.63 ^a^	4.59 ^b^	4.27 ^b^	3.91 ^b^	0.17	0.001

T-AOC = total antioxidant capacity; SOD = superoxide dismutase; MDA = malondialdehyde. ^a,b^ with a row, values with different superscripts are significantly different (*p* < 0.05).

**Table 6 animals-14-00623-t006:** Adonis analysis of the bacterial communities in the cecum and colon of raccoon dogs (*n* = 7).

Item	Cecum		Colon	
	*R* ^2^	*p*-Value	*R* ^2^	*p*-Value
AM0 vs. AM5	0.0971	0.135	0.1776	0.005
AM0 vs. AM10	0.0979	0.181	0.1257	0.031
AM0 vs. AM15	0.1796	0.020	0.2306	0.001
AM5 vs. AM10	0.0970	0.119	0.1813	0.005
AM5 vs. AM15	0.1571	0.020	0.2070	0.004
AM10 vs. AM15	0.1560	0.028	0.1753	0.007

## Data Availability

The data that support the findings of this study are available from the corresponding author upon reasonable request.

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
