# Peer review of "Effects of Dietary Alfalfa Meal Supplementation on the Growth Performance, Nutrient Apparent Digestibility, Serum Parameters, and Intestinal Microbiota of Raccoon Dogs (Nyctereutes procyonoides)"

_animals, 2024, doi:10.3390/ani14040623_

Round 1

Reviewer 1 Report

Comments and Suggestions for Authors

I have read the article titled "Effects of Dietary Alfalfa Meal Supplementation on the Growth Performance, Nutrient Apparent Digestibility, Serum Parameters, and Intestinal Microbiota of Raccoon Dogs (Nyctereutes procyonoides)" and found it to be well-structured and in line with the journal's theme.

The authors investigated the effects of alfalfa meal as a potential supplement to raccoon diets and found that it can alter the composition of the gut microbiota, but does not significantly affect growth parameters.

I have several comments and questions for the authors of the article.

1. I would like to know why the authors chose 5, 10 and 15% concentrations of alfalfa meal in the diet of raccoons, taking into account the specificity of these animals.

2. Whether the health of the animals was monitored during the study. It is known that different concentrations of alfalfa meal in the diet affected the number of Fusobacteria and Treponema in the intestines of raccoons. Treponemes in association with Fusobacteria can lead to various health disorders in animals.

3. In line 236 c - there is an extra letter "T" that should be removed.

4. In line 211, it would be helpful if the authors could clarify what they mean by "Turkey test" or "Tukey test".

5. The authors note that the supplementation of different concentrations of alfalfa meal into the diet of raccoons led to a decrease in ALT and AST activities. However, these changes are not significant.

6. The authors should clarify that these are UCG-003, UCG-005, and UCG-014.

7. To characterize the effect of different concentrations of alfalfa in the diet on the antioxidant system, it is not enough to determine the activity of SOD and malondialdehyde. Why did the authors not study other enzymes and metabolites of the antioxidant system? What is the mechanism of alfalfa meal's effect on the increase in SOD activity and decrease in MDA levels? The authors point to a negative correlation between the number of Fusobacterium and the level of MDA. How can this be explained? Why did the authors not study other enzymes and metabolites of the antioxidant system?

8. The authors contradict themselves in lines 427-430, stating that the changes in SCFA concentration were not significant, but also claiming that they were increasing.

9. It is unclear why the authors determined the content of immunoglobulins in the blood of raccoons.

10. I would like to point out that the conclusion is incorrect in stating that alfalfa meal supplementation improved the antioxidant activity of blood serum, as the total antioxidant activity did not change. Only the activity of one enzyme, superoxide dismutase, and the concentration of one metabolite, malondialdehyde, increased

Author Response

Thank you very much for taking the time to review this manuscript. Please find the detailed responses below and the corresponding revisions highlighted track changes in the re-submitted files.

Comments 1: I would like to know why the authors chose 5, 10 and 15% concentrations of alfalfa meal in the diet of raccoons, taking into account the specificity of these animals.

Response 1: Thank you for pointing this out. The reason behind adding 5%, 10%, and 15% alfalfa meal to the raccoon dogs’ diet was grounded in the literature on alfalfa meal utilization in pig diets. Pigs, being typical omnivores, commonly use the alfalfa meal chosen for this experiment in pig studies as well.

Comments 2: Whether the health of the animals was monitored during the study. It is known that different concentrations of alfalfa meal in the diet affected the number of Fusobacteria and Treponema in the intestines of raccoons. Treponemes in association with Fusobacteria can lead to various health disorders in animals.

Response 2: Thank you for pointing this out. We monitored the animals' health throughout the study and observed abnormal feeding and excretion during daily feedings.

Comments 3: In line 236 c - there is an extra letter "T" that should be removed.

Response 3: Thank you for pointing this out. We have removed the extra letter "T" here.

Comments 4: In line 211, it would be helpful if the authors could clarify what they mean by "Turkey-test" or "Tukey-test".

Response 4: Thank you for pointing this out. We apologize for the misspelling of "Tukey" as "Turkey", which has now been corrected. The "Tukey-test" is one of the multiple comparisons used in one-way analysis of variance when the sample size of each group is the same. In comparison to the LSD test, the Tukey-test is more suitable when there are more than three groups.

Comments 5: The authors note that the supplementation of different concentrations of alfalfa meal into the diet of raccoons led to a decrease in ALT and AST activities. However, these changes are not significant.

Response 5: Thank you for pointing this out. Different levels of dietary alfalfa meal led to a decrease in the serum ALT and AST activities of raccoon dogs, but the effect was not statistically significant. The manuscript also indicates that the p-value is greater than 0.05, and the details can be found on line 266, page 7.

Comments 6: The authors should clarify that these are UCG-003, UCG-005, and UCG-014.

Response 6: Thank you for pointing this out. We are not entirely clear on your meaning. Are you suggesting that Prevotellaceae_UCG-003, UCG_005, and Clostridia_UCG-014 be written as UCG-003, UCG-005, and UCG-014? However, since they belong to different phyla, they should not be abbreviated.

Comments 7: To characterize the effect of different concentrations of alfalfa in the diet on the antioxidant system, it is not enough to determine the activity of SOD and malondialdehyde. Why did the authors not study other enzymes and metabolites of the antioxidant system? What is the mechanism of alfalfa meal's effect on the increase in SOD activity and decrease in MDA levels? The authors point to a negative correlation between the number of Fusobacterium and the level of MDA. How can this be explained? Why did the authors not study other enzymes and metabolites of the antioxidant system?

Response 7: Thank you for pointing this out.

(1) Other enzymes and metabolites of the antioxidant system were also studied during the experiment. However, as there is no kit available for measuring antioxidant indexes specifically designed for raccoon dogs on the market, we utilized a kit designed for dogs. The abnormal data of glutathione-peroxidase (GSH-Px) and catalase (CAT) may be attributed to the fact that the kit was not specially designed for raccoon dogs and cannot be used.

(2) We have not yet explored the mechanism of alfalfa meal's effect on the increase in SOD activity and decrease in MDA level. However, based on other studies on alfalfa meal, it may be related to the bioactive ingredients in alfalfa meal, such as alfalfa flavonoids, alfalfa polysaccharides, and dietary fiber. Further details can be found in the discussion, see lines 485 - 493, page 15.

(3) Spearman correlation analysis revealed a negative correlation between the relative abundance of Fusobacterium and serum MDA content. The results indicate that with the increase of dietary alfalfa meal level, MDA content significantly decreases. However, in the colons of raccoon dogs, the relative abundance of Fusobacterium in the AM5 group is significantly higher than that in the AM0 group.

Comments 8: The authors contradict themselves in lines 427-430, stating that the changes in SCFA concentration were not significant, but also claiming that they were increasing.

Response 8: Thank you for pointing this out. We appreciate your comment, and as a result, we have removed the contradictory section.

Comments 9: It is unclear why the authors determined the content of immunoglobulins in the blood of raccoons.

Response 9: Thank you for pointing this out. During the data review stage before the study, we observed that supplementing animal diets with alfalfa meal could elevate the serum immunoglobulin level in animals. Consequently, we measured the serum immunoglobulin content of raccoon dogs in this experiment. The results revealed no significant effect post-supplementation, contradicting findings from prior studies. However, it does suggest that the addition of alfalfa meal has no adverse impact on the health of raccoon dogs.

Comments 10: I would like to point out that the conclusion is incorrect in stating that alfalfa meal supplementation improved the antioxidant activity of blood serum, as the total antioxidant activity did not change. Only the activity of one enzyme, superoxide dismutase, and the concentration of one metabolite, malondialdehyde, increased.

Response 10: Thank you for pointing this out. Despite the lack of a significant difference in T-AOC among the four groups, T-AOC increased with the escalating alfalfa meal levels. The antioxidant enzyme system, consisting of CAT, SOD, and GSH-Px, plays a crucial role in maintaining low intracellular free radical concentrations. MDA, as the end product of lipid oxidation, serves as a common indicator for determining lipid peroxidation. Although only SOD activity and MDA content were measured in this study, there was a significant increase in SOD activity and a significant decrease in MDA content.

Reviewer 2 Report

Comments and Suggestions for Authors

I feel you have an excellent research project.  However, I have outlined some concerns on the attached PDF.  I believe your research is publishable if you can address theses issues.

Author Response

Thank you very much for taking the time to review this manuscript. Please find the detailed responses below and the corresponding revisions highlighted track changes in the re-submitted files.

Comments 1: Introduction, lines 47 - 50. “Its small, plump body and short legs place it at a disadvantage in the realm of carnivores. Consequently, over the course of evolution, the raccoon dog gradually transitioned into an omnivorous creature.”

Two issues with these statements.

  1. You are stating that the diet of carnivores is primarily meat. This is not true of the canids. The canids are very much omnivorous throughout the family.
  2. The basal carnivores were most likely omnivores. The statement transitioned to omnivorous life style is erroneous.

Response 1: Thank you for pointing this out. We appreciate your comment, and as a result, we have removed the incorrect section, made necessary modifications, and the details can be found on lines 49 - 50, page 2.

Comments 2: “This transformation highlights the raccoon dog's adaptability, allowing it to maintain a diversified diet. The characteristics and functions of the raccoon dog's digestive system are intermediate between those of carnivores and herbivores. The raccoon dog can tolerate roughage, making it suitable for feeding and digesting both animal and plant-based feeds [2]”

Here, you are restating what an omnivore is. As the raccoon dog is a canid, it is a basal carnivore and would be omnivorous from the beginning.

Response 2: Agreed. We have made corresponding modifications to emphasize this point, and the content is the same as above.

Comments 3: Line 82, page 2: What studies. A few of them should be cited.

Response 3: Thank you for pointing this out. We agree with this comment. Therefore, we have included the corresponding references here, please see line 82, page 2.

Comments 4: I understand the need for the 7-day adaptation period. However, for the purpose of the assessment of the microbiota, does the adaptation period introduce a time effect for the introduction of new bacteria.

Response 4: Thank you for pointing this out. The adaptation period will have a time effect on the introduction of new bacteria. However, due to the short duration of the adaptation period, the impact on the formal experiment is minial.

Comments 5: What is chance of the alfalfa meal introducing new bacteria that may affect the outcome of your study?

Response 5: Thank you for pointing this out. Due to the short duration of the adaptation period, it has little influence on the final results of the study.

Comments 6: Line 127, page 4: Define what is considered “normal”.

Response 6: Thank you for pointing this out. "Normal" here refers to the state where food intake and defecation are regular, indicating that the animal is in a healthy condition.

Comments 7: Line 189, page 5: You use a t-test to assess the difference among four groups. Did you correct for multiple comparisons? An ANOVA would be a better statistic to use in this situation. Also, it is confusing as to what is assessed.

Response 7: Thank you for pointing this out. The t-test was employed to evaluate differences between each pair of groups. The assessment focused on analyzing significance differences in species at the phyla and genus levels for the two groups, respectively. Detailed comparison results were presented in Figures S1 and S2 in the supplementary materials. The original statement is unclear, but we have made revisions, please refer to lines 189 - 191, page 5.

Comments 8: Line 211, page 6: Tukey is misspelled as “Turkey.

Response 8: Agreed. We have rectified the error, and the details can be found on line 212, page 6.

Comments 9: Lines 298 - 300, page 8: There were no significant differences in the Simpson index of colonic microbiota among the diets with different levels of alfalfa meal (p > 0.05, Figure 1D).

and lines316 - 318, page 9: Notably, when the level of alfalfa meal supplementation reached 15%, the richness and diversity of the colon's intestinal microbiota in raccoon dogs were significantly increased. These lines appear to be discussing the same assessment, but are stating two different outcomes.

Response 9: Thank you for pointing this out. There are two indices for calculating community diversity, namely the Shannon index and Simpson index. After adding different levels of alfalfa meal to diets, there was no significant difference in the Simpson index of colonic microbiota among groups. However, the Shannon index of the AM15 group was significantly higher than that of the AM0 and AM10 groups, indicating a significant increase in the diversity of colonic microbiota in the AM15 group. These two statements convey the same result and are not contradictory.

Comments 10: Figure 1 shows a great deal of variability in alpha diversity. Is this variability of importance?

Response 10: Thank you for pointing this out. This variability is important. During the evaluation of intestinal microbiota, there were substantial changes in alpha diversity, indicating that the addition of alfalfa meal has a significant effect on the richness and diversity of the intestinal microbiota of raccoon dogs.

Comments 11: Line 438, page 14. You state that physiologic structure constrains monogastric animals efficient utilization of plant-based raw materials. As stated, omnivorous canids might possess ability to digest and utilize these materials to some extent. I agree to some degree. However, the intestinal microbiota aids in digestion of these foods. The coevolution of the microbiota and the animal has allowed the adaptation. Therefore, the intestinal microbiota is substituting for the potential shortcomings of the animal’s anatomy and physiology. Liu et al. (2019) found similar microbiota in the domestic blue fox and raccoon dogs indicating that the microbiota can substitute for the physiology.

Response 11: Thank you for pointing this out. We agree with this comment. Diet is one of the crucial factors affecting intestinal microbiota. Adding alfalfa meal to the diet can alter the intestinal microflora of raccoon dogs, and this microflora can aid in the digestion of alfalfa meal.

Comments 12: Lines 474 and 475, page 15: Urea should not be capitalized

Response 12: Agreed. We have made changes accordingly to emphasize this point, and the details can be found on line 34, page 1; line 153, page 4; lines 267 - 268, page 7; lines 475 - 476, page 15.

Reviewer 3 Report

Comments and Suggestions for Authors

Reviewer’s comments

(1)   The abstract section lacks a description of the results of nutrient apparent digestibility.

(2)   Clarify and enrich the purpose of the study further.

(3)   How to connect raccoon dogs with alfalfa? The significance and practical significance of studying the impact of alfalfa on raccoon dogs need to be further elaborated on.

(4)   What was the reason for choosing 120±5 day old male black raccoon dogs in this study?

(5)   Line 94, briefly describe the source of animals.

(6)   Line 110, table 1, why choose Soybean meal, Wheat bran, and Corn skin to balance the total weight of feeding, that is, why balance the increased weight of alfalfa by reducing the weight of these three?

(7)   Line 232, there is some error in the calculation of F/G in Table 2, please verify all the calculation data in this article by yourself

(8)   Line236, the first line has an additional letter "T".

(9)   Line 257, the measurement of serum indicators in 2.0 materials and methods mentions the determination of IgA, IgG, and IgM, but it is not reflected in the result analysis.

Comments on the Quality of English Language

Minor editing of English language required.

Author Response

Thank you very much for taking the time to review this manuscript. Please find the detailed responses below and the corresponding revisions highlighted track changes in the re-submitted files.

Comments 1: The abstract section lacks a description of the results of nutrient apparent digestibility.

Response 1: Thank you for pointing this out. We agree with this comment. Therefore, we have added a description of the results of nutrient apparent digestibility to the abstract section, and the contents are located on line 33, page 1.

Comments 2: Clarify and enrich the purpose of the study further.

Response 2: Agreed. We have made corresponding changes to emphasize this point, the details can be found on lines 85 - 89, page 2.

Comments 3: How to connect raccoon dogs with alfalfa? The significance and practical significance of studying the impact of alfalfa on raccoon dogs need to be further elaborated on.

Response 3: Thank you for pointing this out. Alfalfa meal is typically used in the diets of ruminants rather than monogastric animals. This is because the insoluble dietary fiber (IDF) contained in alfalfa meal is not easily digestible by monogastric animals. Raccoon dogs, being typical omnivores with a tolerance for coarse feeding, have different levels of alfalfa meal added to their diets in this study. The changes in growth performance, nutrient apparent digestibility, serum parameters, and intestinal microbiota of raccoon dogs were observed to further validate the role of gut microbiota in omnivorous feeding behavior. This study aims to provide a theoretical foundation for the rational utilization of alfalfa meal in the diet of raccoon dogs.

Comments 4: What was the reason for choosing 120 ± 5 days old male black raccoon dogs in this study?

Response 4: Thank you for pointing this out. The rationale for selecting male raccoon dogs aged 120 ± 5 days in this study is that they have entered the late growth period, are relatively mature in growth and development, and possess a certain level of immune capacity. This minimizes the potential impact on the experimental results.

Comments 5: Line 94, briefly describe the source of animals.

Response 5: Agreed. We have briefly described the source of the animals here, and the contents are located on line 95, page 3.

Comments 6: Line 110, table 1, why choose Soybean meal, Wheat bran, and Corn skin to balance the total weight of feeding, that is, why balance the increased weight of alfalfa by reducing the weight of these three?

Response 6: Thank you for pointing this out. As it is necessary to formulate four iso-nitrogen and iso-energy diets, the high crude protein content in alfalfa meal results in an increase in the overall crude protein content of the diet. To maintain the balance of crude protein content among experimental diets, the weight of other protein feeds needs to be reduced.

Comments 7: Line 232, there is some error in the calculation of F/G in Table 2, please verify all the calculation data in this article by yourself.

Response 7: Thank you for pointing this out. The calculation of F/G in Table 2 is accurate. The value filled in the table represent only the average, which comes with a certain standard deviation and standard error. It is not directly calculated from the data in the table.

Comments 8: Line 236, the first line has an additional letter "T".

Response 8: Agreed. We have removed the extra letter "T" here.

Comments 9: Line 257, the measurement of serum indicators in 2.0 materials and methods mentions the determination of IgA, IgG, and IgM, but it is not reflected in the result analysis.

Response 9: Thank you for pointing this out. IgA, IgG, and IgM are serum immune indices, and we have analyzed these three indices in the results. Please refer to lines 276 - 278, page 7. As dietary alfalfa meal supplementation has no significant effect on the serum immune indices of raccoon dogs, we have included the detailed results of this part in the supplementary materials, as shown in Table S1.

Round 2

Reviewer 2 Report

Comments and Suggestions for Authors

Revisions have been made to correct the manuscript to my satisfaction.

Reviewer 3 Report

Comments and Suggestions for Authors

Could be accepted.